# Therapeutic Potential of Mesenchymal Stem Cells in Stroke Treatment

**DOI:** 10.3390/biom15040558

**Published:** 2025-04-10

**Authors:** Mahmood S. Choudhery, Taqdees Arif, Ruhma Mahmood, David T. Harris

**Affiliations:** 1Department of Human Genetics & Molecular Biology, University of Health Sciences, Lahore 54000, Pakistan; ms20031@yahoo.com (M.S.C.); taqdeesarif01@gmail.com (T.A.); 2Allama Iqbal Medical College, Jinnah Hospital, Lahore 54000, Pakistan; ruhma_mahmood@yahoo.com; 3Department of Immunobiology, University of Arizona Health Sciences Biorepository, College of Medicine, University of Arizona, Tucson, AZ 85721, USA

**Keywords:** stroke, stem cell therapy, mesenchymal stem cells, therapeutic potential, regenerative medicine

## Abstract

Stroke occurs when the blood flow to the brain is interrupted due to a rupture of blood vessels or blockage in the brain. It is the major cause of physical disabilities in adulthood. Despite advances in surgical and pharmacological therapy, functional recovery from stroke is limited, affecting quality of life. Stem cell therapy, which may treat neurological disorders associated with brain traumas, including stroke, is an important focus in stroke research and treatment. Stem cell therapy has primarily used a type of adult stem cells called mesenchymal stem cells (MSCs) due to their universality and ability to develop into multiple lineages to regenerate brain cells and repair brain tissues. A significant number of clinical studies provide evidence of the potential of MSCs to treat stroke. This review summarizes the therapeutic mechanism and applications of MSCs in stroke treatment. We also highlight the current challenges and future prospects of adult MSC therapy for stroke treatment.

## 1. Introduction

Stroke is a significant cause of disability and mortality worldwide, marked by a sudden decrease or blockage of blood circulation to the brain. The occurrence of this disruption is typically triggered by the bursting of blood vessels (hemorrhagic stroke) or the formation of blood clots (ischemic stroke) within the brain. Such situations cause the sudden death of the neurons, resulting in persistent damage to the physical and cognitive abilities of the brain. Although stroke prevalence reflects geographical variation, it poses a substantial health burden across all age groups, affecting not only the elderly but also an increasing number of younger individuals [1]. Hemorrhagic strokes, which account for 10–15% of cases, occur when there is bleeding or leakage from blood vessels, resulting in toxic consequences, vessel rupture, and tissue infarction. Ischemic strokes, which account for 87% of all strokes, occur when there is an inadequate flow of oxygen and blood to the brain, usually caused by a blockage in an artery in the brain. The clinical effects of stroke are based on the specific location, nature, and severity of the stroke [2].

Current stroke treatments, such as the use of tissue plasminogen activator (tPA), although effective, are expensive, and their effectiveness depends on the time of infusion after stroke. The prompt administration of tPA after the onset of acute ischemic stroke continues to be a fundamental and essential therapy. The current guidelines advise administering tPA within 4.5 h of the beginning of symptoms, although its administration within 6 h has been considered effective for some patients. Other treatment options include antihypertensive therapy, antiplatelet therapy, neurorepair, and rehabilitation [2,3]. The treatments for hemorrhagic stroke focus on surgical or endovascular procedures or hemostatic therapy to stop bleeding [4,5]. Similarly, the option of surgical removal is not only expensive but also risky. Without proper treatment, rehabilitation can, therefore, not sufficiently address the reversal of the disease. Hence, other options are required for successful rehabilitation and full recovery of lost brain function. Furthermore, the limitations of current treatments highlight the need for innovative approaches that target the underlying mechanisms of brain injury and promote sustainable recovery. Existing therapies for both ischemic and hemorrhagic stroke primarily target the relief of immediate symptoms and the prevention of further decline but do not adequately address the restoration of damaged neural structures for long-term functional recovery. The limited timeframe for treatment, especially with thrombolytic procedures, presents a significant difficulty, as any delays in starting treatment often lead to reduced therapeutic efficacy. The urgent need to dissolve clots and the inability to successfully regenerate the brain tissues highlight the critical importance of prompt intervention for achieving optimal neurological recovery. Given these challenges, stem cell therapy presents a hopeful opportunity to overcome the time constraints and regenerative limitations of current stroke treatments [6,7,8].

With their remarkable ability to differentiate and proliferate into multiple types of cells, stem cells can facilitate the healing and regeneration of tissues and organs. These capabilities make stem cells ideal candidates for use in stroke patients. A promising method of bone tissue regeneration using stem cells for stroke recovery has been adopted successfully [2].

MSCs, a type of adult stem cell, have been widely employed for regenerative treatment options due to their multipotency, superior immunomodulatory abilities, and potential to differentiate into multiple cell types such as neurons, chondrocytes, and osteoblasts [9]. Furthermore, MSCs obtained from adult tissues do not carry the risk of developing tumors that have been observed with totipotent stem cells such as iPSC. The expression of major histocompatibility complex (MHC)-I and MHC-II antigens on MSCs is low, which often eliminates the requirement for immunosuppression after receiving these cells from a different (allogeneic) donor [10]. Due to their wide utility, MSCs are considered one of the most suitable stem cell types for stroke treatment [11]. This review discusses MSC mechanisms of action in stroke treatment, including immunomodulation and anti-inflammatory effects, neuroprotection and neuroregeneration, paracrine signaling, induced angiogenesis, the promotion of neurogenesis, and homing and migration to damaged tissue areas. The clinical studies and therapeutical applications of MSCs in stroke disease are also discussed in detail. The review also highlights the challenges and future advancement of employing MSCs in the treatment of stroke and other neurological diseases.

## 2. Stroke Pathogenesis

A stroke is a sudden disruption of continuous blood supply to the brain, which results in the loss of neurological function [12]. Stroke is categorized into two main types: ischemic stroke, which includes cardioembolic, atherothrombotic, and small vessel disease, and hemorrhagic stroke, which includes intraparenchymal hemorrhage and subarachnoid hemorrhage. Hemorrhagic stroke has a generally poorer prognosis as compared to ischemic stroke, with higher mortality rates, greater likelihood of severe disability, and increased risk of complications [13]. The development of stroke is affected by multiple risk factors, including dyslipidemia, high blood pressure, diabetes, smoking, and atrial fibrillation. Stroke is significantly influenced by blood flow and brain physiology. The brain neurons become deprived of the required energy and oxygen due to the disruption in the blood flow [14]. This leads to the death of brain neurons by apoptosis and the loss of an ATP-dependent intracellular ion concentration gradient. The NF-κB, Notch1, HIF-1α, p53, and Pin1 signaling pathways, which regulate the fate of neurons, are active in ischemic stroke [15]. The outcomes can vary depending on the severity of ischemia. Moderate ischemia leads to the increased expression of genes that promote cell survival. However, severe ischemia and hypoxia trigger the activation of genes responsible for neuronal cell death [14]. Moreover, there is a direct association between the susceptibility to cerebral ischemia and the release of excitatory amino acids (EAAs) outside the cells [16]. Cerebral ischemia is classified based on the level of cerebral blood flow (CBF), which typically ranges from 50 mL to 55 mL per 100 g per minute. From an anatomical perspective, stroke lesions can be categorized into the ischemic central core, which is characterized by irreversible neuronal death, and the ischemia penumbra, which refers to an area with decreased neuronal function but still has the potential to be reversed with acute stroke therapy [17]. Insufficient energy sources rapidly result in the malfunction of energy-dependent ion transport pumps and the depolarization of glia and neurons. The depolarization leads to the emission of excitatory neurotransmitters, primarily glutamate, which worsens the damage by releasing free radicals and disrupting the electron transport chain. Oxidative stress leads to neuronal death by causing damage to the cell membrane. Apoptosis is responsible for the loss of a significant number of neurons, particularly in the penumbra region, in the absence of immediate intervention. Subsequently, astrocytes accumulate around areas of reduced blood flow in the brain and produce proteoglycans that form a glial scar. This scar serves as a barrier, both physically and biochemically, preventing the regrowth and branching of nerve fibers. As a result, it hinders the recovery of neural connections and contributes to the long-term consequences of a stroke [14,17]. Stroke ultimately releases many chemotactic chemicals, including interleukin 8 (IL-8) and monocyte chemo-attractant protein-1 (MCP-1), which attract both stem cells and leukocytes. Specifically, activated endothelial cells produce stromal-derived factor 1a (SDF1a) and its CXC chemokine receptor-4 (CXCR4) following hypoxic injury. Both of them function as chemotactic agents that facilitate the migration of bone marrow and neural stem cells to damaged regions [18]. It is a crucial process for stem cell-based treatments. Neuroprotection therapies have proven ineffective, resulting in inflammation, scarring, and edema. Therefore, the focus has turned towards neurorestorative therapy rather than only preventing further damage. This therapy aims to stimulate the natural growth of new nerve cells, blood vessels, nerve fibers, and connections in the brain tissue by targeting several types of cells, such as neuroblasts, oligodendrocytes, astrocytes, and neurons [19]. Neurorestorative therapies cover a range of treatments, with stem cells being one of them. In addition, there are current research efforts in pharmacology as well as other treatments, including repetitious training, electromagnetic stimulation, constraint-induced therapy, and device-based approaches. Rehabilitation has the potential to utilize the combination of functional reorganization and adaptability following a stroke. Repetitious training focuses on the repetitive practice of specific tasks to improve functional abilities, such as grasping or walking. Electromagnetic stimulation techniques, such as transcranial magnetic stimulation, are a non-invasive brain stimulation method that uses magnetic fields to enhance neural activity and promote neuroplasticity. Device-based approaches, such as neuroprosthetic devices, are artificial devices that replace or support damaged neural functions, such as prosthetic limbs or cochlear implants. Constraint-induced therapy restricts the use of unaffected limbs to encourage the use of affected ones, promoting neural reorganization and motor recovery. Currently, only constraint-induced therapy has proven to have any level of effectiveness [20].

## 3. Major Sources of MSCs

Mesenchymal stem cells (MSCs) are a type of adult stem cell with high regenerative potential. MSCs can be obtained from multiple sources in the body, including adult (such as bone marrow, adipose tissue, dental pulp, etc.) and neonatal sources (such as umbilical cord, Wharton’s jelly, placenta, etc.). Table 1 shows the comparison of adult and neonatal sources of MSCs in terms of their differentiation potential, ease of isolation, potential advantages, and limitations. The choice of source is preferably determined by the ease of isolation and regenerative properties [21]. When choosing a cell source, the potential risks to the donor and difficulties of obtaining the samples must be evaluated. For instance, the isolation of MSCs from bone marrow is an invasive procedure and might lead to hemorrhage, discomfort, or infection. Thus, bone marrow aspiration is more challenging compared to obtaining peripheral blood, birth-derived tissues, or adipose tissue.

### 3.1. Adult Sources

#### 3.1.1. Bone Marrow

Bone marrow-derived mesenchymal stem cells (BM-MSCs) were first identified by Friedenstein in 1976. BM-MSCs became the primary clinical source of multipotent stem cells due to their differentiation capacity, immunosuppressive properties, low immunogenicity, and potential to migrate to sites of injury or inflammation. Nevertheless, obtaining MSCs from humans necessitates a painful and invasive procedure. The regenerative properties of these cells decrease as the age of the donor increases [22]. BM-MSCs have shown significant potential for stroke treatment, as they can differentiate into neural cells, such as astrocytes, neurons, and oligodendrocytes. These cells can replace damaged brain cells. Due to their immunomodulatory effects, they can reduce inflammation and create a conducive environment for recovery. Furthermore, BM-MSCs can migrate to the injured brain area, making them a promising source for stroke patients.

#### 3.1.2. Adipose Tissue

Adipose tissue (AT), alternatively referred to as fatty tissue or fat tissue, is a connective tissue. Adipose tissue is currently the most common source of stem cells. It contains adipose tissue-derived stem cells (ASCs). Adipose tissue is easily available, as it is abundant and readily available. It can be obtained as the byproduct of therapeutic and cosmetic procedures. The functional, phenotypical, and morphological characteristics of ASCs are similar to BM-MSCs. ASCs demonstrate long-term stability in cell cultures, efficient in vitro expansion, and a strong capacity for multilineage differentiation. Interestingly, adipose tissue can be used therapeutically in different forms, including microfat, macrofat, nanofat, stromal vascular fraction (SVF), and as a pure population of ASCs. It is pertinent to note that adipose tissue isolation methods and donor age can impact the therapeutic potential of ASCs [23].

#### 3.1.3. Dental Pulp

Dental pulp is a cluster of fibrous tissue that is present in the middle of the tooth, just below the dentin layer. MSCs can be isolated from the pulp tissue of the third molar. These MSCs have the ability to differentiate into odontoblasts, adipocytes, chondroblasts, and neural lineages. The odontoblasts are responsible for the production of dentin [24,25]. Additionally, dental pulp-derived MSCs (DP-MSCs) have the ability to transform into melanocytes and corneal epithelial cells when grown in a 3D dentin scaffold. A 3D dentin scaffold is a three-dimensional structure used in dental tissue engineering to mimic natural dentin pulp. This ability makes them highly promising for regenerative applications such as the treatment of metabolic disorders and liver disorders such as hepatocellular carcinoma and cirrhosis [26]. Periapical cyst (PCy)-MSCs are a specialized subtype of DP-MSCs that have gained significant interest due to their remarkable ability to proliferate, their unique cell surface marker profile, and the capacity to develop into many different cell lineages, including neurons, osteoblasts, and adipocytes. These cells can be readily obtained from surgically extracted PCys, enabling the recycling of biological waste. These cells also offer a promising alternative for treating neurodegenerative conditions like stroke because of their neural plasticity, which enables them to differentiate into functional neural cells, such as neurons and glial cells, to replace damaged or lost cells [27].

### 3.2. Neonatal Tissues

Neonatal tissues refer to birth-related tissues such as umbilical cord tissue, umbilical cord blood, Wharton’s jelly (WJ), and other birth-associated tissues. These tissues are rich in stem cells, which can treat a range of diseases, including stroke. Stem cells obtained from these tissues have high proliferative potential, the ability to differentiate into multiple types of cells, and a lower risk of immune rejection. Birth-related tissues are abundant and are readily available. They can be cryopreserved for future use at the time of care. One of the key advantages of using neonatal tissue-derived stem cells is that they have a lower risk of immune rejection than other adult tissue-derived stem cells. This ability makes them a favorable option for allogenic use. In addition, the cells obtained from these tissues have high proliferative and differentiation ability. This ability makes them ideal candidates to treat a range of conditions, including blood disorders, tissue damage, and degenerative diseases. The process of the collection and storage of neonatal stem cells, however, is costly and requires specialized facilities. Furthermore, while neonatal stem cells show great promise in preclinical studies, more research is needed to fully understand their long-term efficacy and safety in clinical applications.

## 4. Characteristics of MSCs

Mesenchymal stem cells have many characteristics that make them a promising type of stem cells for many diseases, including neurodegenerative diseases such as stroke. MSCs have the ability to both self-renew and differentiate into many cell lineages. MSCs release various bioactive substances, such as growth factors, cytokines, and chemokines. These growth factors play important roles in the paracrine activities of MSCs [28]. The International Society for Cellular Therapy (ISCT) has defined criteria for characterizing human MSCs. These criteria include (1) the in vitro capacity to adhere to plastic, (2) the expression of CD105, CD73, and CD90 surface markers while lacking expression of CD19, CD11b, CD14, CD34, CD45, and human leukocyte antigen D region (HLA-DR), and (3) the potential to differentiate into adipocytes, chondroblasts, hepatocytes, and osteoblasts in appropriate culture conditions. The most often utilized sources of adult MSCs include umbilical cord, bone marrow, and adipose tissue [9].

The therapeutic efficacy of infused MSCs can be influenced by several factors that affect their quality. As MSCs age, they gradually change in shape and size, becoming flatter and larger (hypertrophy). This change leads to alterations in their capacity to perform regenerative functions, i.e., it declines. For example, the proliferation and differentiation potential of MSCs declines with the advanced age of the donor. Similarly, their immunological features, ability to maintain telomere length, migration to sites of injury, and ability to adhere to other cells significantly decline with in vitro and in vivo aging. Recent research has demonstrated that physiological conditions such as hypoxia preconditioning can increase the regenerative potential of MSCs derived from the elderly. Additionally, the preconditioning strategy can boost the survival of neurons exposed to ischemic stroke. These findings have played a crucial role in advancing the development of effective stem cell-based treatment methods for the elderly.

## 5. Therapeutic Mechanisms of MSC in Stroke Treatment

The fundamental processes of MSC-based stroke treatment are still not entirely understood. Studies have reported mechanisms by which MSCs provide protection against stroke (Figure 1). It is crucial to note that therapies administered immediately after a stroke primarily focus on minimizing the damage, whereas therapies initiated days or weeks later tend to facilitate the healing process. In this section, we discuss the mechanisms of MSCs and the primary proteins released by MSCs that are implicated in the therapy of stroke. These mechanisms include MSC differentiation, the paracrine effect of MSCs, including releasing exosomes, mitochondrial transfer, attenuating inflammation through immunomodulation, induced angiogenesis, promoting neurogenesis, and replacing damaged cells [29,30].

### 5.1. Direct Differentiation of MSCs

MSCs have multipotent characteristics, which allow them to differentiate into many cell types. MSCs are either transplanted directly or naturally migrate to the site of injury or inflammation. Within a particular microenvironment of an organ or tissue, MSCs divide, differentiate, and mature into the same cell type as the tissue or organ. By differentiating into the cells of the respective tissue, they facilitate the process of restoration by replacing damaged cells. This ability enables MSCs to contribute to the regeneration and repair of damaged brain cells and restore their normal function. For instance, MSCs obtained from the human umbilical cord have the ability to transform into cells resembling neurons while also retaining their ability to act as antioxidants and regulate the immune system [31].

### 5.2. Paracrine Effects of MSCs

The paracrine effect refers to signaling biomolecules such as growth factors, cytokines, and hormones produced by cells. These biomolecules subsequently travel short distances to affect surrounding cells and influence their growth, behavior, and function. Recent studies indicate that direct differentiation of MSCs into neural cells may be limited. Infusing MSC-derived conditioned media has been shown to have a similar effect on brain health. Conditioned media contains growth factors, cytokines, chemokines, exosomes, metabolites, and hormones. The conditioned media can be used as a therapeutic agent, mimicking the beneficial effects of MSCs on brain health. The type of culture media used to culture MSCs affects the composition and potency of the paracrine factors, influencing the therapeutic efficacy of the conditioned media. Optimizing media conditions can also enhance the paracrine effects, leading to treatments that are more effective. Thus, the paracrine effect of MSCs has an important role in stroke treatment [31].

The soluble factors produced by MSCs have a significant impact on different types of immune cells, including lymphocytes, dendritic cells, natural killer cells, and macrophages. MSCs produce soluble substances, which contribute to immune regulation and the induction of immune tolerance. Furthermore, MSCs can augment and modulate a negative inflammatory response. Different kinds of paracrine factors form an intricate network of exocrine factors that maintain cellular integrity and promote regeneration. Many tissue-healing models that use MSCs depend significantly on the paracrine function. BM-MSCs have the ability to enhance the production of hepatocyte growth factor [HGF], VEGF, and BDNF, which are derived from astrocytes, in the area surrounding the damaged tissue caused by stroke [32]. These factors have the ability to expedite the healing process by promoting the migration, proliferation, and differentiation of cells. The use of MSCs promotes the production and regeneration of new bone when treating bone injuries like fractures and defects. These research findings provide new treatment strategies for clinical practice, which also stimulate the advancement of regenerative medicine [32]. After stroke, these factors promote the formation of new blood vessels and the recruitment and growth of reactive astrocytes, thereby facilitating the repair of nerve injuries. Furthermore, it has been reported that human BM-MSCs can promote the formation of new blood vessels within stroke lesions by producing natural angiogenic factors that improve the durability of these blood vessels [31]. Hence, the paracrine impact of stem cells is expected to have a significant role in enhancing the density of capillaries and promoting angiogenesis in the injured areas of the brain after stroke [31].

#### Therapeutic Role of MSCs Through Exosomes

Exosomes have an important role in transmitting information and promoting repair through the release of cytokines. The paracrine action exerted by secreted exosomes is important in the process of stroke recovery. Exosomes are lipid particles with a diameter of 40 nm to 160 nm [33]. They are surrounded by two membranes and contain mRNAs and micro RNAs as proteins and lipids. Exosomes exhibit similar therapeutic characteristics as MSCs, such as limited immunogenicity and the capacity to promote nerve and vascular regeneration without any potential for tumor development. Exosomes play a significant role in the development of cell-based acellular therapies for traumatic brain injury, stroke treatment, and other neurological disorders [33]. Research has shown that the direct administration of exosomes could successfully reduce neuroinflammation induced by localized brain injury in ischemic stroke. The extended neuroprotection provided by MSCs-derived exosomes is strongly associated with improved angiogenesis and decreased suppression of the immune system after the ischemic event. This effect created a favorable condition for effective regeneration of the brain. The release of many biologically active substances by MSCs, such as exosomes, is now recognized as the most probable primary mechanism of MSC-based treatment [34].

### 5.3. Mitochondrial Transfer from MSC Therapy

Mitochondria are well-known as the energy powerhouse of the cell because they are responsible for producing adenosine triphosphate (ATP), a powerful substance that powers many biological reactions. Among mitochondria’s important functions are the regulation of energy metabolism, the cell cycle, cell survival and death, apoptosis, ROS formation, and calcium homeostasis [35]. Despite comprising just 2% of total body mass, the brain uses 20% of the energy produced, making it the most energy-intensive organ in the human body. The majority of this energy is utilized for critical central nervous system functions, including the delivery of information through chemical synapses and the generation of action potentials [36]. Neurodegeneration and other disorders may be largely influenced by mitochondrial malfunction. Transplanting healthy stem cell-derived mitochondria to cells damaged by ischemia is a promising new approach to treating ischemic illnesses. Therefore, mitochondrial transfer is an innovative method utilized in stem cell therapy that has gained significant interest [37]. MSCs have the ability to transfer mitochondria to damaged cells that have impaired mitochondrial function. This transfer can occur through several mechanisms and aims to restore the ability of cells to undergo aerobic respiration and improve mitochondrial function. This process helps rescue wounded cells. Mitochondrial abnormalities have been considered as an indication of ischemia injury in these intricate cellular processes [31]. Tseng et al. (2021) showed that the transfer of mitochondria from MSCs to neurons that were damaged by oxidative stress led to improvements in metabolism. The researchers observed the transfer of mitochondria and its beneficial impact on the damaged cerebral microvascular system in rats with cerebral ischemia [38]. Hence, the transfer of mitochondria from MSCs to damaged cells has the potential to provide a new method for treating stroke [38].

### 5.4. MSCs Attenuate Inflammation Through Immunomodulation

MSCs have strong immune-modulatory and anti-inflammatory properties. For example, it has been reported that MSCs can regulate the immune system by suppressing the proliferation of B and T cells, neutrophils, and natural killer cells. MSCs can regulate antibody secretion and synthesis through NK cytotoxicity, B cell activation, and cytokine release. MSCs have been hypothesized to limit monocyte development into dendritic cells and influence their functions. Following a stroke, microglia cells become activated and adopt a phagocytic phenotype, resulting in the production of proinflammatory cytokines. MSCs have the ability to increase the production of anti-inflammatory cytokines, including interleukin-10 (IL-10) and interleukin-4 (IL-4). Alternatively, MSCs have the potential to decrease the production of pro-inflammatory cytokines, such as tumor necrosis factor (TNF), interferon-gamma (IFN), interleukin-1 (IL-1), and membrane cofactor protein-1 (MCP-1). MSCs altered cytokine activity, affecting several immune cell and immunological response pathways, with this ultimately reducing inflammation. TNF and IFN are the principal pro-inflammatory cytokines involved. Prostaglandin E2 (PGE2) is a key mediator released by MSCs, known for its ability to decrease the immune response and inflammation by modifying immunity, limiting T-cell proliferation, and altering T-cell differentiation [39,40].

The level of PGE2 has been shown to drop after stroke. However, MSC transplantation increases the level of PGE2 and decreases the production of TNF in dendritic cells and the production of IFN in natural killer and T helper cells. Subsequently, TNF concentration declined significantly, suggesting that MSCs effectively mitigated neuroinflammation caused by stroke. MSCs not only control the main pro-inflammatory variables discussed earlier but also manage the expression of late pro-inflammatory cytokines, thereby exerting a therapeutic effect. For instance, high mobility group box 1 (HMGB1) is a pro-inflammatory cytokine that is released later in the inflammatory response. Its production can be triggered by TNF and other early pro-inflammatory cytokines. HMGB1 amplifies the inflammatory response induced by initial pro-inflammatory cytokines. The administration of BM-MSCs resulted in a notable decrease in the expression of HMGB1, as well as the receptor for advanced glycation products (RAGE). RAGE, which acts as a receptor for HMGB1, is capable of creating a positive feedback loop that contributes to HMGB1-mediated inflammation. Future research focused on improving the efficiency of MSC-based treatments for neurological conditions will depend significantly on the manipulation of these pathways to control inflammation [41,42].

MSCs play an important role in immunomodulation by regulating CD8+ T cell responses by several mechanisms, such as the inhibition of CD8+ T cell activation and the reduction of neurotoxic CD8+ T cell activity. Excessive activation of CD8+ T cells after stroke contributes to neuroinflammation and secondary brain injury. For example, a study highlighted that high-affinity IL-2 receptor signaling activates CD8+ T cells, which worsens neuroinflammation and causes damage to the white matter of the brain after stroke [43]. Interestingly, another study has reported that human amniotic mesenchymal stromal cells (hAMSCs) suppress CD8+ T cell activation by downregulating IL-2Rα and IL-12Rβ1 signaling in naïve CD8+ T cells. This modulation provides an immunosuppressive environment by reducing cytotoxic potential and proliferation [44].

### 5.5. MSCs Induce Angiogenesis

Angiogenesis refers to the process of generating new blood vessels. This process entails the movement, proliferation, and specialization of endothelial cells into small blood vessels. The vascular density and angiogenic expression in ischemic brain tissue markedly increased following MSC implantation. MSCs showed high expression of factors associated with the density of arteries and growth of new blood vessels. These factors include VEGF, angiogenin-1 (Ang1), tyrosine-protein kinase receptor (Tie2), and VEGF receptor 2 (Flk1). Tie2 is a receptor of Ang1, and Flk1 is a receptor of VEGF. These factors have significant roles in promoting angiogenesis. It has been shown that the introduction of the C-C motif chemokine ligand 2 (CCL2)-overexpressing UCB-MSCs through intravenous infusion resulted in enhanced formation of new blood vessels, primarily due to the migration of these cells towards brain areas that have a greater expression of CCL2 receptors [45]. This migration process further facilitates the natural healing of the brain. This process is mostly achieved by stimulating the release of growth factors and binding to chemokines. Thoroughly studying the underlying mechanisms is critical for preventing the enhancement of symptoms in stroke patients and other undesirable consequences [45,46].

### 5.6. MSCs Promote Neurogenesis

The formation of new neurons in the brain is known as neurogenesis. Ischemic stroke severity is strongly associated with infarct volume size. Adipose tissue, bone marrow, and umbilical cord MSCs were able to decrease the amount of post-stroke infarct, according to in vivo studies conducted on MCAO rats There are a number of factors that can influence the precise impacts on reducing infarct volume following a stroke, including the MSCs’ source, the injection’s timing, the species injected, and the injection dosage [47]. Preclinical stroke trials have shown that MSCs could potentially be able to reduce the severity of post-ischemic motor coordination deficits. Endovascular MSC treatment significantly reduces perifocal vasogenic edema and restores blood-brain barrier function after a stroke [48]. According to preliminary evidence presented by Datta et al. 1 × 10^1^ endovascular MSCs at 6 h post-stroke reduce AQP4 expression and help reduce vasogenic edema, leading to neuroprotection [49].

One method by which MSC transplantation can stimulate neurogenesis is by increasing the proliferation of existing brain cells. For example, BMSC therapy may improve functional recovery after stroke by increasing neuroplasticity and neuronal growth in the ischemia boundary zone (IBZ) [48]. The number of BrdU (+) cells in the surrounding area of the infarct region showed a considerable rise following MSC therapy, suggesting an enhanced proliferation of cells. Undoubtedly, MSCs stimulated the development of axons, synapses, and myelin in the injured brain zone, hence enhancing neuronal function [45]. BM-MSCs markedly enhanced the development of axons in primary cortical neurons as well. Following the administration of MSC therapy to mice with stroke, there was a notable rise in the levels of proteins linked with the formation of axons, whereas the levels of proteins that inhibit axonal growth showed a considerable decrease. The administration of exosomes produced by MSCs resulted in an increase in synaptophysin in the IBZ, leading to an augmentation of synaptic remodeling and plasticity [48].

Another mechanism by which MSC transplantation can facilitate neurogenesis is by protecting nascent cells from their detrimental surroundings, hence avoiding their death. Undoubtedly, the reduction of neuroinflammation results in the prevention of cellular death. A recent study demonstrated the use of MSC spheroid-loaded collagen hydrogels to stimulate the growth of new nerve cells (neurogenesis) and also inhibit the inflammatory response of neurons by producing small, specialized environments that decrease neuroinflammation. The therapeutic effects of MSC spheroid-loaded collagen hydrogels were achieved by enhancing three cell communication signals, subsequently activating a signaling pathway involved in interactions between neuroactive ligands and receptors, and ultimately upregulating the PI3K-Akt signaling pathway. This process led to high expression of proteins associated with neuroprotection and neurogenesis. Facilitating neurogenesis is an essential step in MSC therapy since it allows patients to recuperate after a stroke by repairing injured areas of the brain. Nevertheless, the reparative efficacy of neural stem cells (NSCs) is suboptimal due to their restricted regenerative capacity and the intricate physiological milieu. Therefore, MSCs have the potential to stimulate the transformation of NSCs into neurons by the secretion of various types of nourishing substances (such as growth factors and chemokines) and chemicals (such as nerve growth factor and neuroprotective molecules) that prevent cell death. Subsequent investigations should prioritize the advancement of MSC cell therapies pertaining to NSCs, with the aim of facilitating the restoration of the nervous system [30,40].

### 5.7. MSCs Can Replace Damaged Cells

The transplantation of MSCs can help in the regeneration of brain tissue, primarily via their ability to develop into neurons and glial cells under suitable conditions. Two critical factors are essential for the development of MSCs into neurons, i.e., the expression of nestin and a direct connection between MSCs and neurons, facilitating the incorporation of external inputs. Research conducted over several decades has consistently demonstrated that MSCs possess the ability to undergo differentiation into neurons [9]. Prior research demonstrated the effective induction of MSCs into nestin (+) neurospheres, which were then grown in a medium supplemented with epidermal growth factor (EGF) and basic fibroblast growth factor (bFGF). Following the removal of mitogens from the medium, these neurospheres differentiated into neurons that express neurofilament or glia. When adipose tissue-derived MSCs were differentiated into neurons or glial cells, they contained neuron-specific molecular markers such as microtubule association protein-2 (MAP2), neuron-specific tubulin (Tuj-1), neuron-specific enolase (NSE), and neuronal nuclei (NeuN). These cell surface markers stimulate MSC differentiation into glial and neuron cells to facilitate the regeneration of damaged brain cells. Future research could focus on improving the ways to enhance MSC differentiation into neuronal cells and tissues [50,51].

**Figure 1 biomolecules-15-00558-f001:**
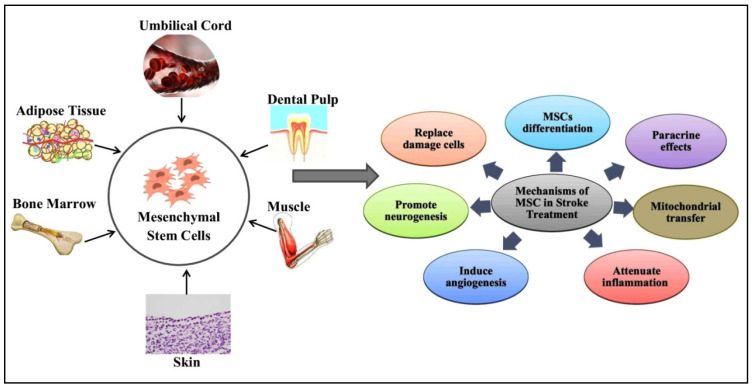
Sources and therapeutic mechanisms of mesenchymal stem cells (MSCs). MSCs can be isolated from the umbilical cord, adipose tissue, bone marrow, skin, muscle, and dental pulp. MSCs can then be employed in stroke treatment. The therapeutic role of MSCs is due to different mechanisms, including direct differentiation into neuronal cells and tissues, paracrine effects, mitochondrial transfer, inflammation attenuation, angiogenesis, neurogenesis, and the removal of damaged brain cells.

## 6. Clinical Studies and Clinical Trials

Several preclinical studies and human clinical trials using mesenchymal stem cells have been registered at www.clinicaltrials.org (accessed on 27 August 2024). These trials either have been completed or are at various stages of completion. The trials are taking advantage of recent advancements in understanding the effects and mechanisms of action of infusing stem cells. A summary of the registered clinical trials was collected from the www.clinicaltrials.gov website on 27 August 2024. The following search was performed: “mesenchymal stem cells therapy for stroke treatment” OR “mesenchymal stem cell treatment” OR “MSCs treatment for stroke”. There are 27 registered clinical trials with different statuses, including not yet recruiting (n = 2), recruiting (n = 8), active, not recruiting (n = 1), completed (n = 7), terminated (n = 0), suspended (n = 2), withdrawn (n = 1), and unknown (n = 6) studies. Table 2 shows some major clinical trials using MSCs for stroke patients. One of the most recent registered clinical trials (NCT06518902), which is planned to be completed in 2025, aims to determine the role of umbilical cord tissue-derived MSCs in treating acute ischemic stroke in adults. The majority of the clinical trials employed adult MSCs. During a 2-year investigation, Gary K Steinberg and his team inserted genetically engineered BM-MSCs called SB623 into brain tissue that had chronic ischemia. The researchers observed that the implantation of SB623 cells in individuals with chronic stroke was both safe and resulted in better clinical outcomes (NCT01287936) [52]. Existing evidence suggests a positive prospect for using different gene targets or preconditioning transformed allogeneic MSCs to treat stroke at all stages. The clinical trials and stem cell research in neurology involved the intravenous administration of autologous MSCs that were cultured and grown using fetal bovine serum. Intravenous autologous MSC transplantation was administered to 15 stroke patients over an extended period of time. The results showed positive outcomes. Since these randomized studies, where the observers were not aware of the treatment given, were not designed to determine effectiveness, it is not possible to draw firm conclusions about the efficacy of this therapy [11,53].

Several recent clinical trials have explored a more focused delivery method, which entails directly injecting or transplanting MSCs into the affected area of the brain, mainly in patients with chronic stroke [54]. The majority of the intravenously administered trials that have been completed are early phase II investigations. In a study, nine patients with serious myocardial infarction (MCA) and NIHSS scores ranging from 10 to 35 were given 2 million/kg body weight of autologous MSCs obtained from bone marrow within 2 months after the initial infarction [55]. When compared to the placebo group, there was no noticeable distinction in neurological healing or functional outcome. Neuroimaging showed some protection against corticospinal tract degeneration, which may support motor function recovery. Still, there was no difference in recovery between the treated and control groups at the 3-month end-point in the STARTING-2 trial, which also included 39 patients with severe MCA who were treated with autologous MSCs (obtained from the bone marrow) [56]. The PISCES-2 study injected 20 million human neural stem cells intra-cerebrally (in the putamen ipsilateral to the infarct) into 23 stroke patients between 2 and 13 months after the initial lesion [57]. A limitation of this study is that it did not include a placebo or non-treated control group. However, it was found that individuals with residual function at the beginning of the study experienced improvements in upper limb function, as measured by the action research arm test (ARAT), which persisted until the 12-month end-point of assessment. For the PASSIoN open-label intervention study, researchers examined perinatal arterial ischemic stroke (PAIS) in humans for the first time. A total of 10 neonates diagnosed with PAIS were given around 50 million BM-derived MSCs intra-nasally in this investigation. A total of 60 percent of patients had an improvement at the 3-month MRI follow-up, with no adverse events (AEs) and raised inflammatory markers [28]. Insufficient preclinical data may predict some failures in clinical trials, as it does not provide solid support for a therapeutic benefit. Prior to conducting efficacy studies, it will probably be necessary to enhance the therapeutic effectiveness of existing stem cell therapies

## 7. Challenges and Future Directions

Despite the exciting promise of MSC-based therapy for stroke treatment, there are numerous challenges and limitations that must be addressed before MSCs can be widely used in clinical settings. The ideal timing for the administration of MSCs is a subject of debate. Presently, the majority of preclinical investigations suggest the transplantation of MSCs within the initial 48 h of an acute stroke is preferred. Studies have indicated that stroke can lead to an elevation in reactive oxygen species, the stimulation of immune cells, and the generation of pro-inflammatory cytokines during the initial stage, hence worsening the subsequent brain damage. Therefore, early transplantation of MSCs should be more effective in stroke treatment. However, the availability of autologous MSCs during this short period of time is a limitation, as the extraction of tissues from a patient during this critical time may pose other risks to patients. Determining the appropriate timing for cell therapy remains a significant problem, as it requires addressing an existing knowledge gap and effectively utilizing existing research. For instance, one study found that exosomes from MSCs have anti-inflammatory and brain-protecting effects in the early stages of stroke [58]. Another study found that introducing MSCs a month after a stroke can still lead to significant recovery of neurological function. The optimal timing for cell therapy in stroke treatment is still unclear, and further research is needed to determine when to administer treatment for maximum benefit.

The optimal source of MSCs for stroke treatment has yet to be identified. While over 90% of preclinical research utilizes freshly obtained MSCs from young and healthy donors, approximately 50% of clinical trials employ autologous MSC products (such as MSC-exosomes and MSC-expanded cells). Autologous MSCs can avoid ethical and logistical issues and have been demonstrated to be more effective than MSCs derived from healthy donors. However, due to the lengthy process of producing an adequate number of stem cells for transplantation, it is not feasible to utilize autologous MSC cells during the acute phase of stroke, particularly in older patients or those with severe illnesses. Reprogrammed or genetically modified MSCs can result in uncontrolled cell growth and genetic abnormalities, which can reduce their survival and therapeutic effectiveness. Moreover, it is yet uncertain whether reprogrammed MSCs can effectively transform into operational brain cells in patients. A comprehensive study found that cryopreserved allogeneic MSCs obtained from healthy donors exhibited low vitality and limited therapeutic effectiveness. Hence, it is crucial to evaluate the survival of allogeneic cells obtained from healthy donors to ensure compatibility with preclinical settings [31,59].

Another challenge is in determining the optimal course of treatment. Clinical studies have demonstrated that MSCs possess immune tolerance and evasion properties in the treatment of stroke. However, preclinical research has shown that the therapeutic effect of exosomes derived from MSCs and conditioned medium can reduce the requirement for actual cells. These cell-free alternative products can be preserved via cryopreservation without any concerns regarding the survival of cells. Cells can be effectively stored for extended periods and easily moved across the globe. However, there is still no agreement on the ideal growth conditions and pretreatment methods to fully optimize the regeneration capabilities of MSC-derived exosomes [60]. The majority of the clinical trials that are phase II have been registered. It is challenging to compare or correlate the studies because of the diverse range of variables, including but not limited to cell delivery time, initial stroke severity, amount of cells supplied, and success of tPA recanalization. The ability of MSCs to cross the blood-brain barrier (BBB) and gradually transfer paracrine substances to the infarcted area of the brain may be a significant constraint. The implications of disrupting the BBB include inflammation after an ischemic stroke, edema, and the disruption of the neurovascular unit. Stem cells cannot cross any barrier in vivo following intravenous or intramuscular injection, but some cells can transiently attach and survive in cerebral capillaries for as long as 72 h [61].

It is pertinent to note that the chances of stroke significantly increase with advanced age. However, the potential of MSCs is negatively regulated with organismal age. As autologous stem cell therapy is required, the use of autologous MSCs from older patients may not be as effective in treating stroke. Therefore, readily available sources of MSCs with reduced immunoreactivity may be required for successful therapy in the elderly [23].

The method of MSC administration poses a significant challenge. Prior research has included systemic and direct methodologies, including intravenous, intra-arterial, and intracranial procedures. More invasive methods (such as intrathecal and intracranial approaches), although possibly causing more harm at the site of injection, may be more effective. Each method has its own advantages and disadvantages. Selecting a simple and safe delivery method for MSCs is a significant challenge in their clinical implementation, demanding increased caution from the clinician [62,63].

Another limitation arises from the diversity in study designs. The lack of methodological consistency in both preclinical and clinical investigations may have significantly contributed to the current inconsistent outcomes. Preclinical research often did not utilize randomized or blinding designs, nor were confirmatory studies conducted, which are necessary for clinical trials. Comparable issues were also present in clinical trials. The studies included randomized controlled trials, single-arm trials, or case series and, therefore, were not able to be directly compared. There is currently no standardized approach for evaluating neurological function, which makes it challenging to come to a consistent outcome regarding the safety and usefulness of MSCs in clinical use. Enhancing methodological consistency is a significant challenge in preclinical and clinical research [31].

The process of aging in MSCs has garnered significant interest in recent years. The passage durations of MSCs in vitro are restricted due to this constraint. Prolonging the duration of growth will result in the occurrence of replicative senescence. In addition, MSCs obtained from older individuals exhibit characteristics associated with aging, which results in their decreased therapeutic efficiency [23]. The presence of comorbidities in patients also poses a challenge for MSC therapy. A significant number of stroke patients have comorbidities such as diabetes, hypertension, and heart disease that can potentially influence the effectiveness of (any) therapy. Medications such as antiplatelet and antidiabetic medicines frequently affect the function of MSCs, hence restricting their therapeutic effects. Unfortunately, the majority of preclinical studies have failed to identify such influential factors, resulting in a significant knowledge gap when it comes to translating stroke research into therapeutic applications [31,64].

## 8. Conclusions

Worldwide, stroke is the most prevalent cerebrovascular disease that results in a significant loss of neurological function. It is also the primary cause of morbidity and mortality. The quality of life of stroke patients is significantly impacted by the limited treatment options available for functional recovery following stroke, despite the progress made in pharmaceutical and surgical interventions. MSCs possess a broad spectrum of possible uses in the management of stroke.

The transplantation of MSCs presents a unique opportunity for the treatment of ischemic stroke. MSCs are involved in numerous pathological processes. They exert their therapeutic effect through various mechanisms, including differentiation, paracrine effects, the stimulation of cell survival, and angiogenesis. However, despite extensive investigation in both preclinical and clinical studies, MSC therapy has not met expectations. The presence of variabilities in cell sources, dosages, dosing intervals, isolation, culture, and expansion techniques pose unresolved challenges. The inconsistent outcomes derived from preclinical research necessitate validation through clinical experiments. It is imperative to conduct a more in-depth investigation into the underlying mechanisms of stroke development and establish suitable animal models that accurately replicate human diseases, taking into account the extent of similarity in neuronal functioning structure, immunology, and metabolism between other animals and humans. However, advancements in the understanding of MSC biology and technological innovations will likely lead to significant advancements in stroke treatment through MSC transplantation and MSC-based acellular therapies in the near future.

## Figures and Tables

**Table 1 biomolecules-15-00558-t001:** Comparison of different sources of mesenchymal stem cells.

Sources	Isolation Method	Potential Advantages	Limitations
Bone marrow	Manual	- Autologous use space - High differentiation capacity into multiple lineages	- Painful and invasive harvesting - Possibility of infection - Number of stem cells is low - Regenerative potential is influenced by the donor’s age
Adipose tissue	Enzymatic digestion	- Preferred source of autologous stem cells - Higher MSC yield compared to bone marrow - Easily accessible and abundant	- Difficult to obtain sufficient quantities from lean and pediatric donors - Possibility of infection - Regenerative potential is influenced by the donor’s age
Dental pulp	Enzymatic digestion, explant culture method	- Rich source of MSCs - Number of colony-forming cells is high - Can be obtained from deciduous (baby) teeth or wisdom teeth Low risk of ethical concerns	- Invasive, as it requires tooth extraction - Less accessible as availability is limited
Birth-derived tissues	Umbilical Cord: Enzymatic digestion, explant culture method Umbilical cord blood: Density gradient Wharten’s jelly: Enzymatic digestion	- Readily available - Non-invasive collection - Low risk of immune rejection	- Expensive equipment for storage - Variable differentiation potential - Uncertainty in long-term efficacy

**Table 2 biomolecules-15-00558-t002:** Clinical trials of using MSCs for stroke treatment.

NCT Number	Study Status	Conditions	Interventions	Phases
NCT06518902	Not yet recruiting	Acute Ischemic Stroke (IS)	UC-MSCs	Phase 1
NCT06129175	Recruiting	Acute IS	Neuron cell axon	Phase 2, Phase 3
NCT05850208	Recruiting	IS	Autologous transplantation of BM-MSCs	Phase 1
NCT05158101	Recruiting	Stroke	UC-MSCs	Phase 1
NCT05008588	Recruiting	IS	UC-MSCs, neurologic and neutrophic drugs	Phase 1, Phase 2
NCT04811651	Recruiting	IS	UC-MSCs	Phase 2
NCT04097652	Recruiting	Acute IS	UC-MSCs	Phase 1
NCT04093336	Recruiting	Infarction, middle cerebral artery, cerebral infarction, infarction, anterior cerebral artery, stroke, brain infarction, ischemic acute stroke,	UC-MSCs	Phase 1, Phase 2
NCT03371329	Completed	Intracerebral hemorrhage, hemorrhagic stroke	BM-MSCs	Phase 1
NCT03356821	Completed	Neonatal stroke, perinatal arterial ischemic stroke	BM-MSCs	Phase 1, Phase 2

Abbreviations: IS: ischemic stroke, UC-MSCs: umbilical cord-derived mesenchymal stem cells, BM-MSCs: bone marrow-derived mesenchymal stem cells.

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
