# Peer review of "Therapeutic Potential of Mesenchymal Stem Cells in Stroke Treatment"

_biomolecules, 2025, doi:10.3390/biom15040558_

Round 1

Reviewer 1 Report

Comments and Suggestions for Authors

This manuscript addresses sources, mechanisms and analysis of preclinical and clinical efficacy of MSCs’ application for therapy of stroke. This topic is relevant to the field because existing clinical therapy is insufficient for quality of recovery for patients with hemorrhagic stroke. There are mainly long-term pharmacological support of patients with stroke for neurological recovery. Many patients do not reach full recovery. MSCs-base therapy gives a chance for patients foe better recovery while starting from the very beginning. This point is very essential.

The article summarizes the data on detailed analysis of source of MSCs (birth-derived tissue is preferential), possible mechanisms of action in stroke ( neurological differentiation, paracrine effects on neuroinflammation, angiogenesis, neurogenesis and neuroprotection, possibilities for MSC-based replacement of damages cells). Existing clinical trials are presented, details of MSCs use for therapeutic purposes are discussed that is very valuable.

The field of study is rather new, therefore  looks attractive. The authors mainly described sources and mechanisms of MSCs beneficial clinical effects in stroke.    In this regard it should be added the information on molecules and pathways of  MSCs -based promotion of neurogenesis and recovery of brain functions.  Sources of MSCs for clinical application were discussed mane times earlier so can be shortened in this review.  Paracrine mechanisms of MSCs promotion of neurogenesis are of special interest. Clinical data are most valuable. So discussion  the protocols of MSCs administration, routes of administration, timepoints for monitoring of clinical and functional parameters of stroke, outcomes and complications of  MSCs therapy may be described more detailed. Results of animal experiments with models of stroke treated by MSCs may be added if possible. Possibility to use primed/neurodifferentiated  MSCs in experimental or clinical studies regarding stroke is also of interest.

I recommended to authors to shortening the description of MSCs sources and to discuss more widely the methodology of MSCs clinical application based on analysis of effective and ineffective outcomes of (models) clinical cases of stroke therapy. Conclusion is rather short due to insufficient of preclinical and clinical data. 

Author Response

This manuscript addresses sources, mechanisms and analysis of preclinical and clinical efficacy of MSCs’ application for therapy of stroke. This topic is relevant to the field because existing clinical therapy is insufficient for quality of recovery for patients with hemorrhagic stroke. There are mainly long-term pharmacological support of patients with stroke for neurological recovery. Many patients do not reach full recovery. MSCs-base therapy gives a chance for patients foe better recovery while starting from the very beginning. This point is very essential.

The article summarizes the data on detailed analysis of source of MSCs (birth-derived tissue is preferential), possible mechanisms of action in stroke ( neurological differentiation, paracrine effects on neuroinflammation, angiogenesis, neurogenesis and neuroprotection, possibilities for MSC-based replacement of damages cells). Existing clinical trials are presented, details of MSCs use for therapeutic purposes are discussed that is very valuable.

The field of study is rather new, therefore  looks attractive. The authors mainly described sources and mechanisms of MSCs beneficial clinical effects in stroke.    In this regard it should be added the information on molecules and pathways of  MSCs -based promotion of neurogenesis and recovery of brain functions. 

Response: Thank you very much for your encouraging comments.

Sources of MSCs for clinical application were discussed mane times earlier so can be shortened in this review.

Response: Thank you for your valuable feedback. We have carefully revised the manuscript to streamline the discussion on MSC sources for clinical applications, reducing redundancy while maintaining essential information. The revised section now focuses on the key aspects relevant to the scope of this review.

Paracrine mechanisms of MSCs promotion of neurogenesis are of special interest. Clinical data are most valuable. So discussion  the protocols of MSCs administration, routes of administration, timepoints for monitoring of clinical and functional parameters of stroke, outcomes and complications of  MSCs therapy may be described more detailed. Results of animal experiments with models of stroke treated by MSCs may be added if possible. Possibility to use primed/neurodifferentiated  MSCs in experimental or clinical studies regarding stroke is also of interest.

I recommended to authors to shortening the description of MSCs sources and to discuss more widely the methodology of MSCs clinical application based on analysis of effective and ineffective outcomes of (models) clinical cases of stroke therapy.

Response: We have expanded the discussion on the MSC-based clinical applications for stroke therapy. Specifically, we have analyzed both effective and ineffective clinical cases to highlight key factors influencing therapeutic outcomes.

Conclusion is rather short due to insufficient of preclinical and clinical data.

Response: Thank you for your observation. We have expanded the conclusion by summarizing all contents of the manuscript.

Response: Considering the reviewer’s suggestions, we have made the necessary revisions, including shortening the description of MSC sources, elaborating on the methodology of MSC clinical applications, analyzing effective and ineffective outcomes in clinical stroke therapy models, and strengthening the conclusion.

Reviewer 2 Report

Comments and Suggestions for Authors

This review is very well written and contains useful information for readers who are specialists in regenerative medicine. However, the review should be deeply revised to make it more attractive and interesting for potential readers.

1- The review contains excessively detailed sections on the mechanisms of development of neurons damage after stroke, as well as types of MSCs and methods of their isolation. These sections should be shortened. In the Stroke Pathogenesis section, it would be better to pay more attention to the mechanisms that stem cells can affect while the general information might be shortened.

  1. The section on types of MSC and methods of their isolation contains information that has been  repeated numerous times in many reviews. It is better to shorten it as well
  2. The manuscript would benefit from the expansion of the main part of the review (sections 5, 6, 7). The sections contain valuable information and there are not as many reviews on this topic as on the previous sections. In the current version, these chapters are written very superficially, there is no in-depth analysis of the problem. There are quite a lot of publications confirming or refuting the differentiation of transplanted cells into neuronal cells, investigating the issues of paracrine regulation and neurotrophic effect of MSCs, and so on. There is really a lot of information on these topics that needs to be understood, analyzed, and summarized, unlike sections 2-4, which contain well-known facts. In addition, it should be noted that sections 5.1, 5.2, 5.3 are actually a retelling of paper number 57 - i.e. they are a retelling of a review.

4 Table 1 contains highly controversial statements. If these are not misprints, the authors should cite literature that proves or explains the validity of the authors' opinion. For example, bone marrow is listed as easily accessible, which is not the case - bone marrow harvesting is one of the most problematic (painful procedure, difficulty in finding material from healthy donors, small portions taken from patients for diagnostic purposes). Perhaps the authors had in mind the lack of a long fermentation procedure? But this is not the main difficulty. Especially since the fermentation of the umbilical cord takes an hour in some techniques, and that of adipose tissue - 30-60 min. Birth tissues, according to Table 1, are difficult to differentiate into adipocytes. This is quite a complicated issue. UCMSCs are easy enough to differentiate into adipose tissue, in our experience we have not noticed any differences from other tissue types. Other authors also confirm our data (10.1155/2013/438243). However, there are authors with a different opinion (https://doi.org/10.1007/s11033-018-4156-1). Other authors believe that the capacity for adipogenic differentiation depends on the part of the umbilical cord from which the cells are isolated (https://doi.org/10.1007/s12015-021-10157-3). In section 3.5. it is desirable to state that it is very difficult to distinguish these cells from dermal fibroblasts and what methods have been proposed to confirm the nature of the isolated cells (MSCs or fibroblasts). In this section it is very desirable to mention the subtlety of the boundary between dermal fibroblasts and dermal MSC.

Author Response

This review is very well written and contains useful information for readers who are specialists in regenerative medicine. However, the review should be deeply revised to make it more attractive and interesting for potential readers.

Response: Thank you very much for your encouraging comments.

1- The review contains excessively detailed sections on the mechanisms of development of neurons damage after stroke, as well as types of MSCs and methods of their isolation.

Response: We have shortened the sections on MSC types, and isolation methods to enhance conciseness while retaining key details relevant to the review's focus.

These sections should be shortened. In the Stroke Pathogenesis section, it would be better to pay more attention to the mechanisms that stem cells can affect while the general information might be shortened.

Response: The authors are grateful for the constructive comments of the reviewer. Regarding the Stroke Pathogenesis section, please see the relevant section (section 5: "Therapeutic Mechanisms of MSCs in Stroke Treatment"). We have now provided more details of the mechanisms through which MSCs exert their therapeutic effects.

  1. The section on types of MSC and methods of their isolation contains information that has been  repeated numerous times in many reviews. It is better to shorten it as well

Response: We have shortened the sections on MSC types, and isolation methods to enhance conciseness while retaining key details relevant to the review's focus.

3. The manuscript would benefit from the expansion of the main part of the review (sections 5, 6, 7). The sections contain valuable information and there are not as many reviews on this topic as on the previous sections. In the current version, these chapters are written very superficially, there is no in-depth analysis of the problem. There are quite a lot of publications confirming or refuting the differentiation of transplanted cells into neuronal cells, investigating the issues of paracrine regulation and neurotrophic effect of MSCs, and so on. There is really a lot of information on these topics that needs to be understood, analyzed, and summarized, unlike sections 2-4, which contain well-known facts. In addition, it should be noted that sections 5.1, 5.2, 5.3 are actually a retelling of paper number 57 - i.e. they are a retelling of a review.

Response: We appreciate your suggestion to expand sections 5, 6, and 7, as they cover critical aspects of MSC-based stroke therapy that require a more in-depth analysis. We have expanded these sections by incorporating a broader range of recent studies.

4 Table 1 contains highly controversial statements. If these are not misprints, the authors should cite literature that proves or explains the validity of the authors' opinion. For example, bone marrow is listed as easily accessible, which is not the case - bone marrow harvesting is one of the most problematic (painful procedure, difficulty in finding material from healthy donors, small portions taken from patients for diagnostic purposes). Perhaps the authors had in mind the lack of a long fermentation procedure? But this is not the main difficulty. Especially since the fermentation of the umbilical cord takes an hour in some techniques, and that of adipose tissue - 30-60 min. Birth tissues, according to Table 1, are difficult to differentiate into adipocytes. This is quite a complicated issue. UCMSCs are easy enough to differentiate into adipose tissue, in our experience we have not noticed any differences from other tissue types. Other authors also confirm our data (10.1155/2013/438243). However, there are authors with a different opinion (https://doi.org/10.1007/s11033-018-4156-1). Other authors believe that the capacity for adipogenic differentiation depends on the part of the umbilical cord from which the cells are isolated (https://doi.org/10.1007/s12015-021-10157-3). In section 3.5. it is desirable to state that it is very difficult to distinguish these cells from dermal fibroblasts and what methods have been proposed to confirm the nature of the isolated cells (MSCs or fibroblasts). In this section it is very desirable to mention the subtlety of the boundary between dermal fibroblasts and dermal MSC.

Response: Thank you for your valuable feedback. We have revised Table 1 to correct misleading statements, acknowledging the invasive nature of bone marrow harvesting and refining the description of adipogenic differentiation potential in UC-MSCs. Additionally, we have clarified Section 3.5 by discussing the challenges in distinguishing dermal MSCs from fibroblasts and outlining methods such as immunophenotyping and differentiation assays.

Reviewer 3 Report

Comments and Suggestions for Authors

Overall Impact

While the review by Choudhery et al., titled "Therapeutic Potential of Mesenchymal Stem Cells in Stroke Treatment," addresses a relevant and timely topic, several areas would benefit from further clarification and development. These include the need for more precise terminology, deeper engagement with recent literature, and a clearer structural distinction between stromal cell subtypes. With these improvements, the manuscript could become a more valuable and informative resource. Therefore, I recommend major revision with substantial restructuring to enhance its clarity, scientific rigor, and alignment with current understanding in the field.

Major Comments

  1. Terminology and Conceptual Accuracy: The manuscript refers to MSCs as "mesenchymal stem cells," a term now largely replaced by "mesenchymal stromal cells" to reflect their limited stemness and predominant function as paracrine mediators rather than regenerative progenitors. This correction is crucial and should be consistently applied throughout the manuscript.
  2. Overemphasis on Regeneration via Differentiation: The review overstates the role of MSC differentiation in tissue regeneration, particularly in the context of stroke. Current consensus, including from Caplan himself, identifies paracrine signaling—not differentiation—as the primary therapeutic mechanism of MSCs. This section requires conceptual revision, with appropriate literature to support the paracrine-centric model (e.g., exosome signaling, trophic factor secretion, immunomodulation).
  3. Need to Distinguish Adult vs. Perinatal MSCs: The manuscript discusses various sources of MSCs without a structured comparison. A clearer classification into adult (e.g., bone marrow, adipose, dental pulp) and perinatal (e.g., umbilical cord, Wharton's jelly, placenta) MSCs is needed. These cell populations differ significantly in their proliferation, immunogenicity, and therapeutic potential. Important work from Ornella Parolini and others on perinatal MSC biology should be cited and discussed.
  4. Insufficient Discussion of Immunomodulation and CD8⁺ T Cells: A major omission is the lack of discussion on how MSCs modulate CD8⁺ T cell responses. Several recent studies outline novel mechanisms:
    • Papait et al., iScience (2023): Downregulation of IL-12Rβ1 and IL-2Rα signaling by hAMSCs in naïve CD8⁺ T cells.
    • Li & Jiang, J Neuroimmune Pharmacol (2025): Role of high-affinity IL-2 receptor in CD8⁺ T-cell-mediated white matter damage post-stroke.
    • Geng et al., Stem Cell Res Ther (2025): Mitochondrial transfer from MSCs to CD8⁺ T cells inhibits cytotoxicity.

These findings constitute a new paradigm in MSC-based immunomodulation and are directly relevant to stroke pathophysiology. Their omission should be addressed.

  1. Inadequate Referencing Throughout: Numerous mechanistic claims are made without proper citation. All assertions—particularly those related to MSC functionality, clinical trial outcomes, and immune mechanisms—must be supported by up-to-date and high-quality references.

Additional Recommendations

  • Include a summary table comparing adult vs. perinatal MSCs.
  • Clarify the limitations of clinical trials (e.g., sample size, controls, endpoints).
  • Emphasize that functional recovery from stroke is linked more to modulation of the immune environment and neuroprotection than to direct cell replacement.

Recommendation: Major Revision

To align with the standards of Biomolecules, this manuscript requires a substantial revision that:

  • Updates terminology to reflect current consensus.
  • Corrects the emphasis on differentiation vs. paracrine mechanisms.
  • Restructures the MSC section by developmental origin.
  • Integrates recent, high-impact literature on immunomodulatory function, especially regarding CD8⁺ T cells.

With these improvements, the manuscript could provide a valuable contribution to the field.

Author Response

Overall Impact

While the review by Choudhery et al., titled "Therapeutic Potential of Mesenchymal Stem Cells in Stroke Treatment," addresses a relevant and timely topic, several areas would benefit from further clarification and development. These include the need for more precise terminology, deeper engagement with recent literature, and a clearer structural distinction between stromal cell subtypes. With these improvements, the manuscript could become a more valuable and informative resource. Therefore, I recommend major revision with substantial restructuring to enhance its clarity, scientific rigor, and alignment with current understanding in the field.

Major Comments

  1. Terminology and Conceptual Accuracy: The manuscript refers to MSCs as "mesenchymal stem cells," a term now largely replaced by "mesenchymal stromal cells" to reflect their limited stemness and predominant function as paracrine mediators rather than regenerative progenitors. This correction is crucial and should be consistently applied throughout the manuscript.

Response: Thank you for your valuable feedback. We acknowledge the evolving terminology and the distinction between mesenchymal stem cells (MSCs) and mesenchymal stromal cells (MSCs), as recommended by the International Society for Cell & Gene Therapy (ISCT). However, the manuscript refers to MSCs as mesenchymal stem cells in the contexts of their stem-like properties.

  1. Overemphasis on Regeneration via Differentiation: The review overstates the role of MSC differentiation in tissue regeneration, particularly in the context of stroke. Current consensus, including from Caplan himself, identifies paracrine signaling—not differentiation—as the primary therapeutic mechanism of MSCs. This section requires conceptual revision, with appropriate literature to support the paracrine-centric model (e.g., exosome signaling, trophic factor secretion, immunomodulation).

Response: After reviewer’s suggestion, we have revised the section to emphasize the paracrine effect of MSCs, incorporating evidence on exosome signaling, trophic factor secretion, and immunomodulation.

  1. Need to Distinguish Adult vs. Perinatal MSCs: The manuscript discusses various sources of MSCs without a structured comparison. A clearer classification into adult (e.g., bone marrow, adipose, dental pulp) and perinatal (e.g., umbilical cord, Wharton's jelly, placenta) MSCs is needed. These cell populations differ significantly in their proliferation, immunogenicity, and therapeutic potential. Important work from Ornella Parolini and others on perinatal MSC biology should be cited and discussed.

Response: Thank you for the suggestion, We have structured the contents as per reviewer suggestion.

  1. Insufficient Discussion of Immunomodulation and CD8⁺ T Cells: A major omission is the lack of discussion on how MSCs modulate CD8⁺ T cell responses. Several recent studies outline novel mechanisms:
    • Papait et al., iScience (2023): Downregulation of IL-12Rβ1 and IL-2Rα signaling by hAMSCs in naïve CD8⁺ T cells.
    • Li & Jiang, J Neuroimmune Pharmacol (2025): Role of high-affinity IL-2 receptor in CD8⁺ T-cell-mediated white matter damage post-stroke.
    • Geng et al., Stem Cell Res Ther (2025): Mitochondrial transfer from MSCs to CD8⁺ T cells inhibits cytotoxicity.

These findings constitute a new paradigm in MSC-based immunomodulation and are directly relevant to stroke pathophysiology. Their omission should be addressed.

Response: Thank you for your insightful suggestions. We have expanded the discussion on MSC-mediated immunomodulation, specifically addressing their effects on CD8⁺ T cells, and incorporated recent studies highlighting novel mechanisms.

  1. Inadequate Referencing Throughout: Numerous mechanistic claims are made without proper citation. All assertions—particularly those related to MSC functionality, clinical trial outcomes, and immune mechanisms—must be supported by up-to-date and high-quality references.

Response: We have carefully reviewed the manuscript and added appropriate citations to support mechanistic claims, particularly those related to MSC functionality, clinical trial outcomes, and immune mechanisms.

Additional Recommendations

  • Include a summary table comparing adult vs. perinatal MSCs.
  • Clarify the limitations of clinical trials (e.g., sample size, controls, endpoints).
  • Emphasize that functional recovery from stroke is linked more to modulation of the immune environment and neuroprotection than to direct cell replacement.

Response: We have addressed the above mentioned suggestions and have incorporated in the manuscript.

Recommendation: Major Revision

To align with the standards of Biomolecules, this manuscript requires a substantial revision that:

  • Updates terminology to reflect current consensus.
  • Corrects the emphasis on differentiation vs. paracrine mechanisms.
  • Restructures the MSC section by developmental origin.
  • Integrates recent, high-impact literature on immunomodulatory function, especially regarding CD8⁺ T cells.

With these improvements, the manuscript could provide a valuable contribution to the field.

Response: We appreciate reviewer’s constructive feedback and suggestions for the improvement of manuscript. We have taken the suggestions positively and have incorporated the required changes.

Round 2

Reviewer 2 Report

Comments and Suggestions for Authors

The authors have significantly improved the MS according to the reviewer's comments.  Thank you.  However, the text still requires language editing

Comments on the Quality of English Language

The quality of the English should be improved before publication.  There are numerous linguistic errors ("age" instead of "go into senescence", "as" instead of "as well as", punctuation, etc.).

Reviewer 3 Report

Comments and Suggestions for Authors

The manuscript has now been substantially improved and is suitable for publication